# Understanding Barriers to Health Behaviours in 13–17-Year-Olds: A Whole Systems Approach in the Context of Obesity

**DOI:** 10.3390/nu17081312

**Published:** 2025-04-10

**Authors:** Helen Lambert, Barbara Engel, Kathryn Hart, Jane Ogden, Katy Penfold

**Affiliations:** 1School of Biosciences, University of Surrey, Guildford GU2 7XH, Surrey, UK; h.lambert@surrey.ac.uk (H.L.); b.engel@surrey.ac.uk (B.E.); k.hart@surrey.ac.uk (K.H.); 2School of Psychology, University of Surrey, Guildford GU2 7XH, Surrey, UK; klpenfold@hotmail.com

**Keywords:** young people, obesity, whole systems approach (WSA), diet, activity

## Abstract

Background/Objectives: This study examined factors influencing health behaviours among 13–17-year-olds in Surrey, focusing on rising obesity rates and socioeconomic disparities using a whole systems approach to capture both the stakeholders’ voice and the young people’s voices. Methods: The research involved two components: a survey of youth service providers (e.g., teachers, youth workers; *n* = 35) and focus groups with adolescents (*n* = 27). Results: The survey revealed challenges faced by stakeholders, including insufficient training, environmental factors (e.g., schools, social media, food systems), and limited support from parents and healthcare professionals. The focus groups identified two key themes: (1) domains of care, for example diet and how availability and cost of food affects food choices, and (2) barriers and solutions, addressing financial, structural, and emotional obstacles to and facilitators of a healthy lifestyle. Transcending these themes was the key role of health inequalities linked to income, geography, and gender. Conclusions: This study underscores the complexity of adolescent health behaviours and calls for a multi-level, coordinated approach to address inequalities and foster supportive environments for healthier choices.

## 1. Introduction

Nearly two-thirds (64%) of adults in England are either overweight or living with obesity (BMI ≥ 25 kg/m^2^) [1], while, in Surrey, this figure is 59%. Research consistently shows that children who are overweight face a heightened risk of becoming overweight or obese adults, accompanied by serious health and social implications [2].

The National Child Measurement Programme (NCMP) is an essential component of the Government’s Child Obesity Plan, which sets out strategies to “significantly reduce childhood obesity” [3]. However—as with the rest of England—the societal distribution of weight status is unequal and strongly associated with level of deprivation. There is a 19% difference in obesity prevalence at Year 6 between the most and least deprived children nationally [1], and, within Surrey, the overall statistics mask a wide variation in obesity prevalence across the county, with levels highest in areas of greater child poverty and deprivation.

The immediate determinants of weight are physical activity and diet, though these are the outcomes of many inter-related underlying causes. Less than half of children and young people in Surrey aged 5–16 meet the Chief Medical Officer’s recommendations of 60 min of physical activity a day, which is similar to the figure for England as a whole (47%) [1]. Many young people have a sedentary lifestyle, with almost half of 15-year-olds in the UK reporting a mean sedentary time of over 7 h per weekday [4]. Young people (11–18-year-olds) in the UK consume 12.3% of their total energy as sugar, more than double the recommended maximum of 5%, and have the highest intakes of sugar-sweetened drinks of any age group (142 g/day). Only 12% of this age group meet the 5 A Day recommendation for fruit and vegetable intake [5]. Adolescence is a critical period for the development of obesity and obesity during teenage years increases the likelihood of persistent obesity in adulthood along with the associated health risks [6]. In England, modelling of data obtained from the National Household Survey has suggested that over 13% of poor diet nationally could be linked back to adverse experiences in childhood [7]. Accordingly, this is a time when health promotion could play a particularly important role.

During this vulnerable period of physical and emotional development, many young people report that mental health is their primary concern [8], with more than 20% of 11–16-year-olds reporting a probable mental health disorder [9]. The relationship between mental health and eating habits is complex, particularly in adolescence, and often cyclical in nature, affecting and being affected by factors including self-esteem and anxiety and associated with disordered eating, trauma, and depression [10,11]. Body dissatisfaction is also common in adolescents [12]. For example, only half of young people surveyed in the What About Youth (WAY) survey [4] thought they were the “right size” at age 15. Parents exert a formative influence on health-related behaviours, particularly dietary habits [13,14], but parental influence decreases during adolescence [15], with peers, role models, and social media becoming stronger influences [16]. This makes adolescence a particularly vulnerable time for young people with respect to establishing and maintaining healthy behaviours but also provides a window of opportunity during which these can, with appropriate intervention, be formed and influenced.

Recent systematic reviews confirm the association between healthy dietary behaviour and/or sufficient activity levels and better psychological wellbeing, with many relationships reported for factors such as sleep, stress, diet, and physical activity with each other and with body weight [17,18,19]. From these reviews, it is clear that health behaviours such as diet, activity, and sleep influence wellbeing in the broadest sense including factors such as wellbeing, mood, body image, and mental health. Such factors are also linked to body weight. The direction of such relationships, however, is less clear, often due to the designs of the studies used and the possibility of third factors or reverse causality. It is often concluded, therefore, that these relationships are likely to be bi-directional and often mediated and moderated by a wealth of other factors.

Adolescence is thus a critical life stage for shaping future health, marked by a complex interplay of factors influencing weight status, and physical and mental wellbeing. Despite this, there is a notable lack of targeted strategies or interventions for this age group, in contrast to the abundance of programs for younger children [20,21]. Adolescents are often perceived as “hard to reach”, further compounding the challenge of addressing their unique needs. A holistic (whole system and evidence-based approach) to improving health in this age group is urgently required.

The Office for Health Improvement and Disparities (OHID), formerly Public Health England (PHE), commissioned its Whole Systems Approach (WSA) to Obesity programme in 2019 as a response to the growing understanding of the complexity of tackling overweight and obesity, and the relative ineffectiveness of previous approaches [4]. Systems-based approaches take into account the structure and function of different systems and how they interact with each other. Using WSA for obesity has shown promise across both adults and children in a number of countries [22,23,24]. In the UK, several counties, including Surrey, have successfully implemented such approaches for a range of public health issues including obesity, in both adults and children [25,26,27]. Within Surrey, there is currently provision for addressing overweight in younger children, including the ongoing Be Your Best programme [28], which targets children of 0–12 years and their families. However, presently, there is a gap in provision for 13–17-year-olds.

A programme of work was therefore designed and undertaken, applying the WSA to identify whether similar initiatives could be developed for young people in Surrey.

The aims of this project were as follows: (i) to understand what factors influence health lifestyle choices for children aged 13–17 in Surrey; (ii) to identify the key elements which should be considered in developing an effective action plan to support children aged 13–17 to make healthier food choices in the County of Surrey and thus support attainment of a healthy weight.

The programme followed the four main phases of the WSA defined in the Public Health England recommendations [23,24].

This paper reports on the findings from Phase 2 of the research programme, which included a survey of stakeholders working with children aged 13 to 17 in Surrey, alongside a focus group study involving adolescents in the same age range. The aim of Phase 2 of a WSA is to “build the local picture” [24], and it was therefore important to gather the experience of both these key stakeholder groups. Each study is presented separately, and the results are synthesised in the discussion section for a comprehensive analysis.

## 2. Study 1: The Stakeholders’ Voice: A Survey

### 2.1. Method

#### Design

A single-group mixed methods questionnaire design was used to explore stakeholders’ experiences of working with children aged 13–17 in supporting them to make healthy lifestyle choices, including barriers and facilitators. This design integrated both quantitative and qualitative approaches to capture a comprehensive understanding of the stakeholders’ perspectives. Data were collected between June 2023 and March 2024.

-Quantitative Component

The quantitative aspect of the questionnaire included 9 Likert scale items to assess the ways in which stakeholders interacted with young people with respect to healthy eating, physical activity, weight management, mental health, and wellbeing, exploring the nature of these interactions, stakeholder confidence, and barriers to addressing these issues within their role.

-Qualitative Component

The qualitative component consisted of open-ended questions within the same questionnaire, where participants were invited to elaborate on specific challenges they faced, strategies they found successful, and any contextual factors that they felt influenced their work. This allowed for a more in-depth exploration of individual experiences and facilitated the identification of nuanced barriers and facilitators not captured by the quantitative items. The open-ended questions are shown in Table 1.

### 2.2. Participants

Key stakeholders were identified in consultation with Surrey County Council and Active Surrey. This included Surrey County Council funded groups as well as independent groups from a wide remit encompassing theatre, dance, sports, Scouts, YMCA, youth clubs, disability support, and faith groups. Contact details were provided to the research team by Surrey County Council so that invitations to participate in the survey (and focus groups) could be cascaded through these contacts.

### 2.3. Procedure

The questionnaire consisted of 14 closed questions, 9 of which were Likert-scale, relating to the participants’ interactions with young people with respect to five areas: healthy eating, physical activity, weight management, mental health, and wellbeing. This was followed by 10 open-ended questions about the challenges faced in supporting young people and what would help them provide this support within their roles. Prior to completing the questionnaire, participants were required to provide their informed consent within the survey tool. The survey was conducted anonymously and took approximately 10 min to complete.

### 2.4. Data Analysis

Likert scale responses were analysed within Qualtrics. Qualitative data were analysed using the six-step framework method of thematic analysis [29], involving (i) familiarisation (researchers read the transcripts multiple times to ensure familiarisation with the content); (ii) generating initial codes (sections relevant to the aim of the study were colour-coded and categorised, leading to the identification of patterns among the data); (iii) searching for themes (those with similar content were grouped together to create a 15-theme hierarchy); (iv) reviewing the themes (refining the themes and codes after discussion to ensure they were distinct and relevant enough for an evidence-based and clear narrative; (v) defining and naming the themes (the themes and subthemes were illustrated in a thematic map, and relevant extracts from the transcripts were used to develop a thematic story); and (vi) producing the report (writing the report, enabling some iteration between the themes and the selected quotes).

### 2.5. Results

The participants were 35 stakeholders, with the majority working in either education (22%) or youth work (20%). Some respondents did not specify their role (27%).

Likert scales: Participants reported that their most frequent interactions within their roles related to wellbeing and mental health, with the least frequent being around weight management. More than a third of respondents (34%) had daily interactions regarding wellbeing, and over a quarter (26%) reported daily interactions around mental health. In contrast, only one respondent reported daily interactions about weight management and 43% reported that they never dealt with this topic. The majority of interactions relating to health (30–50%) arose in the context of an activity, although some reported raising the issue themselves (32% for wellbeing compared to 16% for healthy eating or weight management) or that the young person raised the issue (36% for mental health, compared to 10% for healthy eating or physical activity).

The majority of respondents felt confident discussing mental health and wellbeing, healthy eating, and physical activity with their young people. However, less than half (36%) were confident discussing weight management, and 49% reported that they were somewhat unconfident or extremely unconfident to do so.

Only 17% of respondents had received any training in the area of weight management, compared to 80% for wellbeing and 77% for mental health. The figures for healthy eating and physical activity training were 46% and 43%, respectively. This was emphasised by responses to questions regarding what people thought would enable them to feel better supported in their roles, with 76% stating that more training would enable them to better discuss weight management with the young people in their service.

Other barriers to addressing health issues included “not wanting to cause unintentional harm” (most common reason cited for weight management and mental health), “out of my remit/not my role” (most common reason for physical activity and healthy eating), and “lack of time” (most common for wellbeing). Overall, the majority of respondents wanted to be able to do more to address all of these health issues with their young people.

Free-text responses: Participants’ free text responses highlighted a number of challenges to supporting young people. The results are presented in Table 2.

Overall, the qualitative results highlighted key roles for a lack of knowledge, with a focus on their lack of training; the environment, with an emphasis on schools, the government, social media, and the food environment; and issues with support, such as parental support as well as external support including nutritionists/dietitians and GPs and the need for better access and clearer signposting.

## 3. Study 2: The Young Person’s Voice: Focus Groups

### 3.1. Participants

The inclusion criteria for this study were as follows: aged 12–18, speak and understand English, and live in Surrey. Participants (*N* = 27) were 14 females and 13 males, age 12–16 years (12-year-olds were included if they were in school year 8). Organisations working with young people were invited to participate in the study as detailed above for study 1, and further recruitment was undertaken by contacting organisations whose details were available in the public domain. Those who agreed to participate then acted as gatekeepers for the purposes of recruiting young people to the focus groups, obtaining informed consent from the young people and their parents, and organising the meeting within which the focus group would be held. A total of four focus groups were conducted, of which three took place in youth groups, organised by the youth leaders, and one in a school, organised by members of the teaching staff.

### 3.2. Design

A qualitative design was used to obtain detailed data sufficient for a rich and in-depth understanding of the experiences of young people aged 13–17 years. Data were collected using focus groups, and were analysed using thematic analysis following the method outlined by Braun and Clarke [29]. The data collection process took place between June 2023 and March 2024.

### 3.3. Procedure

The focus groups consisted of 6–8 participants and were facilitated by two members of the research team, in the presence of the youth leader or teacher for that group. The sessions lasted approximately 40 min and included discussions on healthy eating, physical activity, weight, mental health, and wellbeing. Young people were prompted to share whether these were areas they thought about, where they obtained advice and information, and what the barriers and facilitators of these aspects of health were.

Participants were invited to use pseudonyms during the discussion, which was recorded using an encrypted digital Philips recorder issued by the University of Surrey. Written notes were also made to facilitate subsequent transcription.

### 3.4. Data Analysis

Recordings were transcribed, anonymised, and analysed in NVivo (version 14) using thematic analysis [29]. The final themes were then interpreted in relation to the research question, offering insight into the participants’ perspectives.

## 4. Results

Analysis described two main themes relating to “Domains of care”, which highlighted the areas of their health young people deemed to be important, and “Barriers and solutions” to maintaining their health, which focused on financial, structural/physical, social, and emotional factors. Transcending these themes was the key role of “Health Inequalities”, specifically relating to money, geography, and gender. These themes are illustrated in a thematic map (see Figure 1).

### 4.1. Theme 1: Domains of Care

The analysis revealed several key domains that participants identified as central to managing their health. These domains highlight the varied factors that teenagers consider important for maintaining their wellbeing, extending beyond just physical health to include mental and emotional aspects. Three primary subthemes emerged: diet and exercise, body weight, and mental health and wellbeing.

(i)Diet and exercise

Participants frequently mentioned diet and exercise as integral components of their health management. Many teenagers expressed an awareness of the importance of maintaining a balanced diet and engaging in regular physical activity to support overall health. However, several factors influenced their dietary choices and exercise habits, including personal preferences, family routines, and social influences. For instance, some participants described how their home environment influenced their eating habits, such as not having healthy food available:


*“So, if you’ve got healthy things in the fridge, you eat healthy, and if not, you’re hungry, so you eat what’s in there.”*


In one case, a participant described how his step-father influenced his behaviour:


*“My stepdad honestly he gives me too many sweets when I’m, whenever I’m playing games on my xx. So I mean, like he would probably be the worst reason why I eat sweets and chocolate the worst.”*


In terms of exercise, one focus group said that there was plenty of sport available to them and so were happy and had no further suggestions. For the others, a number of reasons for low physical activity were described, including time spent in sedentary activities (TV, video games, time on phones); psychological factors, such as a lack of motivation, stress at home, and poor mental health; physical factors, such as bad sleep schedules, neurological disorders, not being strong enough, or being injured; and environmental factors, including the cost of activities, the weather (too hot or too cold), and negative school influences (“bad” teachers, overly competitive sessions, group allocations). Young people reported being too “cool” for sport, having less opportunity to participate as they got older, and receiving insufficient encouragement from their parents, Gender was also identified as a prominent issue affecting participants’ comfort and confidence in engaging in sports. Female participants specifically described experiencing sport-related sexism, which contributed to their reluctance to participate. This discomfort was often rooted in fear of judgment from peers, teachers, and coaches. However, one participant noted that the perception of sexism and the associated discomfort could vary significantly depending on the quality of the teacher or coach, as expressed in their reflection:


*“As long as your coach is good and supports you it doesn’t matter your gender as long as they know what they are doing and support you, encourage you.”*


This perspective underscores the importance of positive role models in fostering an inclusive environment in sports, highlighting that effective support can mitigate the impact of gender-related barriers.

(ii)Body weight

Body weight emerged as a significant focus of health management for teenagers, encompassing not only concerns about physical appearance but the associated impact on self-esteem and social acceptance. Participants often expressed worries about body confidence, including feeling “too fat”, “too thin”, “too tall”, or “too short”:


*“I feel insecure about my weight and like…How big my belly is, but really like nothing ever gets said about it. So, I don’t know if that’s actually that bad.”*



*“The thing is you can’t. You can’t be too skinny because then you could be called anorexic. But you can’t be too fat because you’d be called, you’d you’d be called fat. But then at the same time, you can’t want to be skinny.”*


These insecurities were largely created by bullying at school, as well as online content (social media) from celebrities and influencers, and the unrealistic nature of photos (i.e., by using Photoshop):


*“The internet, it’s so bad with like body image and how you look, what you eat…makes you feel so bad about yourself when you’re trying your hardest and you see all these people with perfect bodies.”*


Concerns about body weight and its impact on participation in sports were also highlighted by several participants, particularly regarding their interactions with coaches. One individual recounted feeling shamed by their sports coach for gaining weight and emphasised the pressure to conform to specific body expectations. They reflected on the perceived consequences of weight gain, stating:


*“If I was really overweight, there’d be a warning like, ‘We’ll drop you if you don’t lose weight.’ So, then I’ll just go to them and say, ’Set me a diet then’.”*


Another participant echoed similar sentiments, suggesting that gaining weight could jeopardise their position on a team. They articulated the belief that coaches prioritise athletic performance, stating:


*“Oh, cause then if you’re like severely obese and you play for a club, they’re not gonna want to play you because you won’t be the best player if you’re not fit. Yeah. So, then you’d be like, ‘How do you want me to get fit?’ And then they could just take you on fitness plans and a diet.”*


These accounts illustrate the pressure some young people feel regarding their weight and fitness, revealing how external expectations from coaches can lead to feelings of shame and anxiety about body image.

(iii)Mental health and wellbeing

Outside of school, as with the subtheme of body weight, online content was the main source of mental health problems (via influencers and Photoshop); however, in school, it was academic stress and the pressure to do well. Participants also described being shouted at by teachers when they did not understand the work, unenjoyable school experiences, and bullying as key sources of mental health problems in school.


*“Like I said, all they [the teachers] were saying is ‘ohh just ignore them. [the bullies] Try to not do anything with them and like, try stay away from them’, but it’s really hard to do that, yeah and like you can say that, but that’s that’s never ever gonna help and also, people if you’re getting angry or sad, people might say ‘ohh take a chill pill, calm down’, but that never actually helps. Yeah they might not. They might not know that, but it literally never helps so I mean, I’m just saying that.”*



*“I think school as a whole can make people stress a lot like if you get give them too much and work. Then you could get too stressed or if you get if you have a test or a quiz that you’re not quite prepared for soon, then you might be like ‘ohh, what if this happens? What if that happens?’ Well, if I get a bad grade what if my parents are disappointed in me or something? And then like all the stress just builds up from every single bit of School and then like you have to release it some way, so it might also come out as you becoming a bully at school or being a bad person at school.”*


These quotations demonstrate not only how academic stress can affect people’s wellbeing, but that it can also feed into the cycle of bullying. A range of places they would go for mental health were identified, including the doctor, head of school, head of year, and helplines (like Samaritans), although generally young people felt that support from schools was lacking.


*“The thing is she [the school counsellor]. She’s like. I don’t trust her anymore because she left me in her room for half an hour with the door wide open by myself. Like. Literally crying my eyes out and she just left me like that. You don’t gain students trust from it. You gain like you don’t gain anything from it and there is like your job here is to be a supporter, to a student, but you’re not supporting them at all.”*


In contrast, at one centre, participants felt as though mental health was being spoken about too much, that they were saturated with it and that talking about it all the time made things worse as they focused on it more.


*“I feel like sometimes it is like, oh, there’s this and this that could happen. Like, if you’re feeling slightly anxious then and it’s sometimes I feel like it is overdone a little bit.”*


The quotation suggests that discussions about mental health in schools, while important, might sometimes unintentionally amplify feelings of anxiety. For example, individuals who are only mildly anxious may be exposed to a wide range of potential mental health issues during such discussions, which could feel overwhelming or excessive to them. The above quotations demonstrate the complex nature of mental health, and the difficulty schools and pupils face when trying to tackle it.

### 4.2. Theme 2: Barriers and Solutions

The analysis revealed significant barriers to healthy lifestyle choices among teenagers aged 13–17 in Surrey, as well as potential solutions to mitigate these challenges. The subthemes are financial barriers and solutions, structural/physical barriers and solutions, social barriers and solutions, and emotional barriers and solutions.

(i)Financial barriers and solutions

Financial barriers to health and wellbeing included the cost of food in schools, i.e., less healthy food is often cheaper than more nutritious, healthy food, making it difficult for children to make healthier choices. Specifically, they highlighted that food in schools was often more expensive than less healthy options which they could purchase outside of school.


*“I could get a full Tesco meal Deal for it and still have money left over. But in school you get like half a slice of bread.”*



*“School food is really overpriced.”*


Participants also highlighted the expense of sports clubs, gym memberships, and clothing/kit for sport as significant barriers. They reported that teachers and other service providers discussed financial struggles in terms of obtaining kit for sport, indicating that financial barriers are pervasive in Surrey, not just on an individual level but on a group level:


*“Like at times you can’t do much because you don’t have the money to or it’s too expensive. Like there’s so many opportunities that so many people want to do. But it’s too expensive because like they obviously can’t really drop the price because they’ve got to earn a living off of it. But at the same time for people to go, it needs to be cheaper because it only seems like the people who have the have the amount of money that could go.”*


Some of the focus groups also described the lack of available sporting facilities in some areas, or that sporting facilities were run-down. However, one group felt that the facilities both in their schools and surrounding areas were satisfactory, which might highlight the impact of financial inequality in different areas of Surrey.

(ii)Structural barriers and solutions

Participants expressed frustration over the limited availability of healthy meal options in schools. Many reported that school meals predominantly consisted of unhealthy choices, contributing to poor dietary habits and compounded by easy access to junk food. Respondents noted that unhealthy snacks and fast-food outlets were readily available in various environments, making it difficult to opt for healthier alternatives. A participant said:


*“Sometimes the school meals I feel are a bit like these aren’t very healthy.”*



*“They’re [school dinners] not great. The quality of the food is really not brilliant so it’s like all terrible.”*


The study also identified challenges related to participation in sports. Participants mentioned the lack of sports programmes and accessible facilities, which hindered their ability to engage in physical activities. Additionally, participants indicated that a limited variation in sports offered in schools led to disengagement, especially for those who did not enjoy traditional sports like football. A participant reflected:


*“If you don’t like football, there’s really nothing else to do. It would be nice to have more options.”*


Safety concerns were also noted, particularly when crossing roads to access parks or sports facilities. Several participants mentioned the lack of adequate street lighting, which deterred them from participating in physical activities. A 14-year-old remarked:


*“It’s scary walking to the park at night. I’d rather stay home than risk crossing dark streets.”*


A lack of awareness regarding available sports and activities further compounded these barriers. Many teenagers reported insufficient information about local opportunities for physical activity. One participant expressed:


*“I wish I knew more about what sports I could join. I feel like there are a lot of options, but I just don’t know about them.”*


Furthermore, some respondents highlighted a lack of support within schools for mental health issues, including long waiting lists for Child and Adolescent Mental Health Services (CAHMS). They described inadequate resources and the perceived ineffectiveness of school counsellors:


*“But even mental health service like CAMHS, like CAMHS in horrendous. You’re on the waiting list for two years, unless you’re, like, severely, severely depressed, suicidal. And they put you on the like the red zone. And as soon as they think. Ohh yeah, you’re good. They’ll take you off it and put you back on the waiting list and never see you again. And like they just don’t get you help. It’s not good.”*


The cleanliness of their town was also a concern for participants, who linked the state of their environment to mental wellbeing. A participant mentioned:


*“When I walk around and see litter everywhere, it just makes me feel stressed and unhappy. It’s like no one cares.”*


In summary, the barriers identified in this subtheme underscore a multifaceted landscape affecting teenagers’ health management, with structural obstacles that necessitate targeted solutions to foster healthier lifestyle choices.

(iii)Social and emotional barriers and solutions

Emotional factors were found to significantly influence teenagers’ ability to make healthy lifestyle choices. The subtheme of emotional barriers encompassed various psychological and social challenges that hindered their efforts to engage in healthier behaviours.

Linked to the previous subtheme, domains of care, bullying was identified as a significant emotional barrier, particularly in the context of physical activities. Participants recounted experiences of being ridiculed by both peers and, in some cases, adult supervisors when they struggled with sports. This negative social feedback led to avoidance of sports participation. One person said:


*“I makes you not want to do it, cause if they’re rude to you and don’t encourage you then you just want to give up on the sport and leave.”*


This negative feedback also included sexism, with girls often feeling unwelcome due to their gender. Many participants also reported difficulties in expressing their feelings or seeking support, often due to a perceived lack of trustworthy adults or peers. This emotional isolation contributed to feelings of helplessness and reluctance to seek help. Similarly, fear of judgment or stigma associated with seeking assistance emerged as a barrier to accessing support. Participants described feelings of embarrassment when considering reaching out for help, which often prevented them from taking the first step.


*“You get made fun of for asking for help when you need it.”*



*“You would be really embarrassed, and they would tease you about that and then you might become the bully.”*


Although not a direct barrier, the reliance on CAHMS highlighted a gap in emotional support for body confidence and self-esteem. Participants indicated that the limited availability of timely mental health resources made it challenging to address issues related to body image or motivation to pursue healthy habits. A participant noted:


*“CAHMS is great when you can get help, but the waiting lists are so long. It’s like the support isn’t there when you actually need it.”*


Overall, these emotional barriers highlight the need for comprehensive support systems that address not only physical health but also the psychological factors influencing teenagers’ health behaviours. Addressing these emotional challenges could improve the likelihood of adolescents adopting healthier lifestyle choices.

### 4.3. Transcending Theme: Health Inequality

Transcending these themes was the theme of health inequality, which existed on different strata including money, geography, and gender.

Financial challenges disproportionately affect teenagers’ ability to adopt and sustain healthy habits. Limited financial resources create barriers to accessing nutritious food, sports equipment, and recreational opportunities. Many participants shared that healthier school meals were often unaffordable, with cheaper junk food outside school becoming a tempting alternative. The cost of participating in sports—ranging from club fees to gym memberships and the necessary equipment—emerged as a frequent obstacle.


*“Yeah, like big families and all that. Like at times you can’t do much because you don’t have the money to or it’s too expensive. Like there’s so many opportunities that so many people want to do. But it’s too expensive because they obviously can’t really drop the price because they’ve got to earn a living off of it. But at the same time for people to go, it needs to be cheaper because it only seems like the people who have the have the amount of money that couldn’t go.”*


Service providers echoed these concerns, underscoring the difficulties of offering affordable programmes and resources. Financial inequality not only restricts access to healthy food but also curtails opportunities to engage in physical activity, further entrenching the cycle of health inequity.

Closely linked to financial inequalities, geographical inequalities was a recurring theme, with participants from more affluent areas reporting better access to sports facilities and opportunities, while those in less affluent areas highlighted significant challenges. Teenagers in wealthier areas expressed satisfaction with the availability and quality of local sports programmes and facilities. In contrast, participants from less well-off areas frequently cited run-down or non-existent sports facilities, inadequate amenities such as goalposts in parks, and generally unsafe environments. For example, when asked what would help improve the health of people in one of the less affluent areas, several of the participants stated more lighting, as they felt unsafe going to the park:


*“More lights around, like the park area because it gets like pitch it’s pitch black and you have to like walk through the park.”*



*“More lights cause I know on my street like this lady’s car got vandalised like a couple of times. But I think it’s been so dark, but because the lights turn off so early, they can’t actually see who’s doing it on the cameras.”*


These disparities were compounded by financial constraints, as the high cost of equipment and participation further restricted access to available options. These geographical disparities underline how critical it is to address the uneven distribution of resources to ensure all teenagers have the same opportunities for physical and mental wellbeing.

In terms of gender, participants identified sexism as a significant barrier to participating in sports, with many expressing that being female often made them feel unwelcome or discouraged from engaging in athletic activities. This sexism was multifaceted, manifesting in various ways, such as societal beauty standards. For example, girls reported being told by boys that they looked “manly” if they played sports or faced criticism for developing muscles, which challenged traditional notions of femininity.


*“Boys are so horrible to anybody else that plays sport. They’re always like, they’re always but they’re always like, oh, you look like a man or you’re a man for doing that. They’re like, take the mick out of you for playing a boys sport. If it’s a boy dominated sport, they’re like you’re not like you’re a lesbian.”*


The above quotation also additionally shows that this sexism sometimes intersected with discriminatory attitudes toward sexuality. One participant recounted girls being labelled as “lesbians” in a derogatory manner simply for participating in sports. This not only underscores the pervasive sexism they face but also reveals underlying homophobia, where the term “lesbian” is weaponised as an insult.

## 5. Discussion

The growing prevalence of obesity among adolescents in England, particularly in areas like Surrey, is a significant public health concern, with marked socioeconomic disparities contributing to the inequitable distribution of obesity across different regions [1]. This two-part study aimed to understand the factors influencing health behaviours in adolescents aged 13–17 in Surrey, and to identify key elements for developing an effective action plan to promote healthier lifestyle choices.

### 5.1. Study 1: The Stakeholders’ Voice

The findings from this survey highlight several critical challenges and opportunities for stakeholders working with young people aged 13–17 to support healthy lifestyle choices. The quantitative results reveal that stakeholders are most confident discussing mental health and wellbeing, followed by healthy eating and physical activity, but feel less confident about addressing weight management. This is consistent with the low levels of training reported by stakeholders, with less than 20% of respondents receiving training in weight management, and the expressed desire for more training in this area. The qualitative data further elaborate on these challenges, with stakeholders reiterating that a lack of knowledge and training are major barriers to effectively addressing health issues. Many also noted that the environment, including the school and food environments, and external support systems, such as parental and professional involvement, could be more supportive. These responses underscore the complexity of supporting healthy lifestyle choices in young people, where factors beyond the stakeholders’ control, such as social media influences, inadequate resources, and a lack of clear pathways for accessing support, play significant roles.

Despite these challenges, stakeholders reported a strong commitment to improving their practice. They identified several facilitators which would help them to do this, such as better signposting to available services and more structured pathways for support. The findings suggest that a more integrated approach, combining training for stakeholders with better resource allocation and a clearer support system for young people, could help bridge these gaps. Furthermore, the study points to the need for systemic changes, such as healthier school environments and stronger parental engagement, to create a more supportive ecosystem for promoting healthy lifestyles among adolescents. This reflects recent reviews of the research indicating that multicomponent interventions involving parents, teachers, and digital involvement offer “a promising strategy” for promoting healthier behaviours and weight management in young people [20,21]

Given the complexity of the factors influencing stakeholders’ abilities to support young people, these findings suggest that targeted interventions at both the individual and systemic levels are essential for improving outcomes in this area.

### 5.2. Study 2: Young People’s Voice

The qualitative findings from this study provide valuable insight into the complex factors influencing healthy lifestyle choices among adolescents aged 13–17 in Surrey. As the prevalence of obesity continues to rise, particularly in areas of socioeconomic disadvantage [1], it is crucial to understand not only the direct behaviours related to diet and physical activity but also the underlying psychosocial and environmental influences that shape these behaviours and support or prevent them from changing. Through the use of focus groups, the study identified two key themes that offer insights into the “domains of care”, and “barriers and solutions” adolescents face when making healthy lifestyle choices.

“Domains of care” highlighted the areas which young people felt were most important to them and focused on their diet, body weight and mental health. In terms of diet, several factors influencing choices were identified. Adolescents from more affluent backgrounds reported better access to healthier food options, while those from deprived areas described challenges in obtaining fresh, affordable foods, an economic disparity that has been previously recognised [1]. Whilst family support continued to play a significant role in shaping adolescents’ eating habits, with parents serving as the primary decision-makers in food provisioning, reflecting previous research [13,14], the strong influence of peers and social media was also apparent. The participants in this study revealed that peer pressure significantly influenced their food choices, particularly in social settings such as school or gatherings with friends. Additionally, they noted that social media had a dual impact: while it could promote unhealthy body ideals and contribute to body dissatisfaction, it also served as a source of positive health messages, offering inspiration for fitness and healthy eating. These findings suggest that while social media can contribute to negative body image, it also holds potential as a platform for health promotion and for encouraging positive lifestyle behaviours. The discussion on social media’s dual influence on adolescents’ health behaviours, both as a source of inspiration and as a contributor to body dissatisfaction, relates directly to recent policies aimed at regulating or banning social media for young people. Policymakers have increasingly scrutinised the impact of social media on adolescents’ mental health and wellbeing, leading to actions aimed at restricting or modifying its accessibility [30]. However, this study highlights that social media can also serve as a positive force. While outright bans may seem more straightforward than strategies like moderating content or promoting media literacy, they risk overlooking the potential benefits social media offers.

The study also explored the role of body weight in adolescents’ health behaviours, particularly focusing on the relationships between body dissatisfaction and dietary habits. Many participants reported feeling dissatisfied with their body image, with body weight being a source of concern, a common occurrence in this age group [11,12,19]. Such issues have been associated with unhealthy coping mechanisms, including poor dietary choices and low physical activity levels [11,12,19]. However, participants highlighted several solutions including better access to sporting facilities/equipment, incentives to participate in sport such as vouchers or monetary incentives, availability of healthier food more cheaply in schools, and a better understanding of nutrition and food preparation. Future research could explore how implementing these factors affects body weight and health more broadly.

The results of this study also highlight the significant role that academic stress, bullying, and online content play in shaping the mental health and wellbeing of teenagers in Surrey. Many participants reported that the pressure to excel academically was a major source of stress, exacerbated by negative interactions with teachers, which is reflective of previous research [31]. Additionally, bullying within the school environment contributed to feelings of anxiety and lowered self-esteem. In terms of seeking support, participants indicated various outlets including doctors, school staff, and helplines like Samaritans. However, there was a general feeling that schools were not offering sufficient mental health support, which echoes national data [6,11,19,31]. Interestingly, while some participants appreciated discussions around mental health, others felt overwhelmed and believed that focusing on it too much only made the situation worse. This highlights the complexity of addressing mental health in educational settings, where a balanced approach is necessary to avoid exacerbating students’ concerns. These findings emphasise the need for a more holistic, nuanced approach to mental health support in schools, one that addresses both external stressors and the emotional experiences of young people and which can be tailored to reflect individual needs.

In terms of “barriers and solutions”, participants identified a number of structural barriers, and participants in more deprived areas reported being affected by the lack of safe outdoor spaces and often faced pressure to stay indoors, either due to concerns about safety or a lack of resources for extracurricular activities. This aligns with previous studies indicating that neighbourhood deprivation can play a significant role in shaping health behaviours [2]. It underscores the importance of interventions aimed at improving access to healthier environments, particularly in highly deprived areas. Such interventions should address both macro-level societal changes (i.e., societal change) and micro-level individual support (i.e., improving individual environments), fostering environments that promote health and wellbeing.

Financial constraints were a significant barrier to adopting healthier eating habits, particularly for adolescents from lower socioeconomic backgrounds. Participants in the study reported that healthier food options were often more expensive and less accessible than cheaper, less nutritious alternatives. The financial barrier highlights the importance of policy-level interventions aimed at reducing the cost of healthy foods for low-income families.

The emotional barriers to adopting healthy lifestyle habits were also significant. In line with previous research [11,12,19] (and as highlighted above), body dissatisfaction was a key emotional trigger for unhealthy eating behaviours. Adolescents with low self-esteem often felt overwhelmed by the pressures of social media and societal expectations, which exacerbated their emotional struggles and unhealthy eating patterns.

Participants in the study identified several solutions to the barriers they face. In terms of financial solutions, participants suggested that subsidised programmes or financial assistance could help make healthier food choices more affordable. Whilst research is lacking, this strategy has been shown to be successful in improving family diets. In a multi-method study exploring whether food vouchers can improve nutrition and reduce health inequalities, participants reported that the vouchers increased the range of fruit and vegetables they used and improved family diets, establishing good habits for the future [32]. However, children in receipt of free school meals are likely to experience stigma and bullying by peers, which may hinder efforts to encourage healthier behaviours [33,34,35]. Future research could explore alternative strategies to motivate young people towards healthier lifestyles.

Regarding structural solutions, participants highlighted the importance of schools in promoting healthy behaviours, suggesting that schools should offer healthier meal choices and improve physical activity programmes. They also emphasised the need for more accessible community resources for physical activity, such as local sports clubs and safe outdoor recreation groups, to encourage young people to be more active. Emotionally, participants highlighted the importance of positive role models and peer support, to motivate healthier behaviours. They also recognised the need for increased access to mental health support, as mental health issues, such as anxiety and depression, were often linked to unhealthy eating behaviours. Addressing body image dissatisfaction through education on body positivity and self-esteem was also seen as a critical solution to overcoming emotional barriers. Finally, while parental influence decreases during adolescence [13,14], participants suggested that families should be supported with resources and education to make healthier choices together, providing a strong foundation for long-term health behaviour change. Together, these solutions reflect a holistic approach to overcoming the barriers to health behaviours and promoting healthier lifestyles for 13–17-year-olds in Surrey.

The transcending theme “health inequalities” describes the intersection of financial, geographical, and gender inequalities, highlighting the complex, layered nature of health disparities among teenagers in Surrey. The experiences of health inequality are not singular but are shaped by multiple, intersecting factors that require nuanced interventions. Prior research has demonstrated that childhood obesity is strongly associated with level of deprivation [1]; the results of this study highlight such disparities, and call for policies that address these interconnected barriers, such as subsidies for sports programmes in underserved areas; initiatives aimed at challenging societal beauty standards; and increased investment in safe, accessible public spaces for physical activity. Additionally, given the gendered nature of problems such as weight stigma, body dissatisfaction, and mental health [36,37,38], tackling gendered and homophobic stereotypes in sports should be a priority, as these attitudes not only limit opportunities for girls but also perpetuate broader societal inequities.

Ultimately, addressing these health inequalities requires a comprehensive approach that considers the financial, geographical, and gendered dimensions of access to health-promoting resources. Accordingly, when both individual and structural changes are made, the individual can feel empowered to make better choices and develop their sense of autonomy whilst being supported by the world around them [16]. Without such a holistic approach, the cycle of health inequity will persist, disproportionately affecting those who are already marginalised and limiting their opportunities for a healthy future.

### 5.3. Limitations

A mixed-methods design was chosen because it allowed for both the breadth of understanding afforded by quantitative data and the depth of insight provided by qualitative responses. The quantitative data offered a structured overview of stakeholders’ experiences, while the qualitative responses provided context and elaboration, giving a fuller picture of the challenges and successes in supporting children aged 13–17 to make healthy lifestyle choices.

The studies presented, while providing valuable insights into adolescent health behaviours, are not without limitations. Firstly, the sample sizes and demographics may limit the generalisability of the results. Both studies were conducted in Surrey, which may not be representative of adolescents from different geographic locations, socioeconomic backgrounds, or cultural contexts. Additionally, the recruitment process through gatekeepers such as youth leaders and teachers could introduce selection biases, as the participants may have been influenced by the characteristics of the organisations they were recruited from.

While focus groups offer valuable qualitative insights, they are subject to group dynamics that may affect individual responses. The presence of facilitators, such as teachers or youth leaders, could also introduce bias and influence the comfort level of participants when discussing sensitive topics such as body image and mental health.

Both studies are cross-sectional, meaning they capture a snapshot of adolescent health behaviours at a single point in time. Longitudinal studies, which track changes in health behaviours over time, could provide a more comprehensive understanding of how adolescent health evolves and how external factors (such as social media influence, family dynamics, or policy changes) impact health behaviours over time. Additionally, the focus of both studies on specific health domains—diet, exercise, body image, and mental health—limits the broader understanding of adolescent health. Other important factors, such as sleep, substance use, and environmental influences, were not explored, which may have provided a more holistic understanding of the health challenges faced by adolescents.

Future research would benefit from expanding the sample size, exploring diverse populations, using longitudinal designs, and addressing a wider range of health factors to provide a more comprehensive understanding of adolescent health.

Finally, there are some problems inherent within the WSA approach that need to be considered. Primarily, whilst stakeholder engagement throughout the process is key to bringing about effective transformation [22,23,24,39], the constraints on resources and competing priorities faced by public sector and voluntary organisations represent fundamental barriers to adopting such an approach [23]. Accordingly, stakeholders may offer insights into solutions but without funding these solutions cannot be implemented. Further, implementing evidence-based and effective interventions based upon a WSA require the evaluation of the WSA. This requires the need for ongoing collaboration between academia and public health decision-makers [23,39], which itself requires additional funding and raises discussions about whether funding should be spent on the delivery or evaluation of these approaches. Finally, the WSA is a complex and multifaceted approach which generates complex and multifaceted solutions which, whilst more meaningful, are inherently more difficult to implement than more simplistic ones.

## 6. Conclusions

In conclusion, despite their awareness of the importance of balanced diets, physical activity, and mental wellbeing, adolescents’ attempts to maintain healthy behaviours are hindered by financial, structural, and social and emotional challenges, including limited access to nutritious food, inadequate physical activity resources, and bullying. Likewise, stakeholders working with young people, such as teachers and youth group workers, also face barriers due to insufficient training and a lack of integrated support systems. The present study therefore emphasises the need for a coordinated, multi-level approach involving families, schools, and communities to create supportive environments that facilitate healthier choices. This approach should address financial issues, such as the expense of sport and equipment; structural issues, such as the availability of healthy food and safe physical activity spaces; and emotional concerns, like body image dissatisfaction and mental health support. The study also highlighted various health inequalities, such as money, geography, and gender, indicating that targeted interventions, cognisant of these differences, are warranted. A holistic approach is essential to break the cycle of inequity and support marginalised groups. Reducing these barriers would make it easier for teenagers to make sustainable health decisions, promoting overall wellbeing and resilience.

## Figures and Tables

**Figure 1 nutrients-17-01312-f001:**
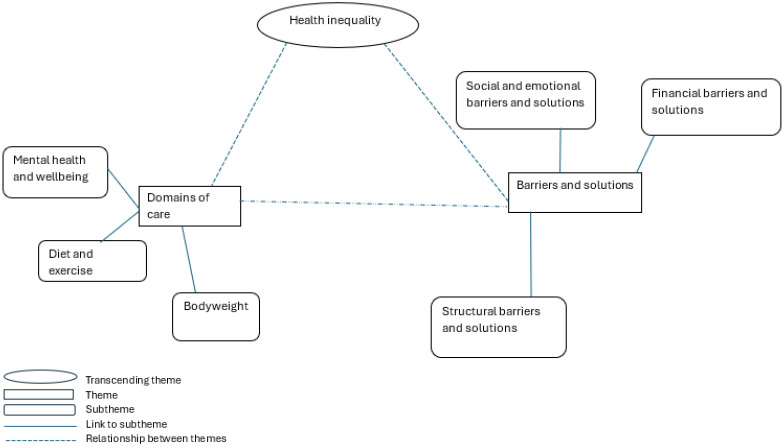
Thematic map representing relationships between areas of health deemed important by young people and barriers and solutions to health.

**Table 1 nutrients-17-01312-t001:** Open ended questions for stakeholder survey.

Topic	Items
Healthy eating	What challenges do you face in supporting young people to make healthier eating choices? What would help you to support young people with making healthier eating choices better?
Physical activity	What challenges do you face in supporting young people to be physically active? What would help you support young people with their physical activity better?
Weight management	What challenges do you face in supporting young people with weight management? What would help you support young people with weight management better?
Mental health	What challenges do you face in supporting young people with their mental health? What would help you support young people with their mental health better?
Wellbeing	What challenges do you face in supporting young people with their wellbeing? What would help you support young people with their wellbeing better?
General	Can you think of anything else that might help you support young people’s health?

**Table 2 nutrients-17-01312-t002:** Thematic analysis of free-text survey responses provided in an online survey by adults working with young people in Surrey.

Theme	Subthemes Explored	Verbatim Example	Freq of Theme	Theme Totals
Knowledge	Lack of training (youth leader)	“Education establishments to offer more training opportunities for staff to feel more confident on delivering certain topics”	30 (22%)	37 (28%)
	Lack of knowledge (youth leader)	“Would like to be knowledgeable” “Lack of knowledge on how to support weight management without causing harm”	7 (5%)	
Environment	School Environment	“Better/healthier options in the canteen” “More varied meals in the canteen”	5 (4%)	20 (15%)
	Food Environment (availability, price)	“What is readily available to young people (e.g., energy drinks, cheap processed foods)” “Food on offer on-site” “Increased availability and lower costs of healthier foods”	9 (7%)	
	Social media	“Social media and marketing companies pushing unhealthy food on our children” “Screens, internet and social media provide unhealthy habits and content”	2 (1.5%)	
	Government (NHS)	“Little support from NHS” “More joined up work between local NHS services i.e., dieticians, GP surgeries” “Culture of eating ultra-processed foods is endemic, and requires policy change and governmental intervention”	4 (3%)	
Support	Parental support	“Require parents to provide a variety of alternatives”	3 (2%)	22 (16%)
	External support (nutritionists/GPs)	“More training/education from experts for the kids” “Talks in school” “External speakers and workshops for young people” “Input from dietitian specialists in college—a drop in, or tutorial session for students, a training session for staff”	8 (6%)	
	Pathway of support	“More access to signposting” “An easier/central way of finding out what is available in the local area and how to access these services” “A lack of clear pathway and process for school nurses to follow” “A clear pathway to follow … when to refer to other services” “Better knowledge of available services in the areas I’m working”	11 (8%)	
Resources	Lack of financial resources	“More funding for our catering team to put on regular healthy eating days throughout the year” “More funding for events and training” “Sufficient funding to support small groups on a regular and recurring basis” “More funding for activity opportunities” “More incentives for teenagers to continue with the physical activity”	7 (5%)	36 (27%)
	Lack of practical resources	“Short, snappy resources to hand out” “Readily available resources” “Basic visual resources” “Limited age-appropriate resources” “Better websites/agencies to signpost to”	20 (15%)	
Education	Lack of education for young people	“There is a lack of education as to what ‘healthy’ eating is” “A drop-in or tutorial session for students” “Earlier education at school” “Ability for more children to understand why activity is good for you”	15 (11%)	19 (14%)
	Lack of education for parents	“Education to families as to the benefits of physical activity, eating healthy and sleep.” “More parent education classes/workshops to improve their knowledge” “Too many options and families who don’t understand the benefits of eating healthy”	4 (3%)	

## Data Availability

The original contributions presented in the study are included in the article, further inquiries can be directed to the corresponding author.

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
