# Peer review of "Understanding Barriers to Health Behaviours in 13–17-Year-Olds: A Whole Systems Approach in the Context of Obesity"

_nutrients, 2025, doi:10.3390/nu17081312_

Round 1

Reviewer 1 Report

Comments and Suggestions for Authors

This manuscript provides a thorough and well-structured analysis of adolescent health behaviors, with a particular focus on obesity, socioeconomic disparities, and the Whole Systems Approach (WSA). The study effectively integrates both quantitative and qualitative data, offering valuable insights into the barriers and facilitators influencing young people's health choices.

  • Clarify the Intersection Between Mental Health and Obesity: The manuscript discusses body dissatisfaction, bullying, and academic stress, but the causal pathways between mental health and obesity could be better articulated. For example, how does chronic stress influence eating behaviors? Could emotional eating be a mediating factor? More specific psychological mechanisms would strengthen this section
  • Expand the Discussion on Gender Disparities: The study acknowledges gendered differences in health behaviors but does not fully explore their implications. Are boys and girls equally affected by weight stigma and social media influences? Could gender-specific interventions (e.g., tailored physical activity programs) be beneficial?
  • Provide More Depth on the Role of Schools in Health Promotion: Schools are mentioned as critical environments for shaping health behaviors, but what specific interventions have been successful in similar contexts? Could school meal programs, curriculum changes, or mental health initiatives be incorporated into future recommendations?
  • Address the Potential Limitations of WSA: While the WSA is promoted as an effective strategy, its challenges (e.g., cost, stakeholder coordination, sustainability) are not discussed. Are there barriers to implementation that policymakers should consider?
  • Refine the Conclusion for Greater Impact: The conclusion effectively summarizes key findings, but the policy recommendations could be more concrete.

This manuscript provides a valuable contribution to the field of adolescent health and public health policy. Strengthening the discussion of mental health, gender differences, school-based interventions, and policy recommendations would enhance the manuscript’s depth and practical applicability.

I recommend minor revisions to address these areas, but overall, this is a high-quality and impactful study.

Reviewer 2 Report

Comments and Suggestions for Authors

Dear Redactors,
Thank you very much for the opportunity to revise the article “Understanding Barriers to Health Behaviours in 13-17 Year Olds: A Whole Systems Approach in the Context of Obesity”.
This study examined factors influencing health behaviours among 13–17-year olds in Surrey, focusing on rising obesity rates and socioeconomic disparities using a Whole Systems Approach to capture both the stakeholders’ voice and the young people’s voices. 
The research involved two components: a survey of youth service providers and focus groups with adolescents. The survey revealed challenges faced by stakeholders, including insufficient training, environmental factors, and limited support from parents and healthcare professionals. The focus groups identified two key themes:domains of care, for example diet and how availability and cost of food affects food choices, and barriers and solutions, addressing financial, structural, and emotional obstacles to and facilitators of a healthy lifestyle. 
Unfortunately, I have the feeling that the research does not solve the problem and indicate factors that are well-known and obvious. The question is how to solve the problem of obesity among adolescent and this article didn’t show possible explainations. 
Additionally, the study groups are very small, which make the conclusions unreliable. 

Thanks.

Reviewer 3 Report

Comments and Suggestions for Authors

The paper is really interesting in its contents, and it discussed important domains for adolescents and the role of stakeholders.

The paper is well written even if it is really very long because it showed results of two studies, one on stakeholders qualitative and quantitative information, and another on adolescents’ experiences.

I have only some minor comments to ameliorate this paper.

Introduction

Try to make some tables or figures to explain the indices of the different parameters in England. I think that you can cut a little in this part that is really very verbose and not strictly related to the next shown studies.

Introduce more the reasons for the two developed studies, it is not clear.

Method

Quantitative questionnaire isn’t showed deeply, and it should be explained more in detail (line 158). Then in the text it is described the qualitative questions, but not this part.

What about informed consent given to adolescent participants and to their parents? What about the authorization for the stakeholders?

Results

The qualitative approach is interesting, but it is really very long and the reader can lose the main idea of the study.

Try to shorten this part or try to make some tables or figures about this part.

Discussion

I suggest to add that a training is necessary for stakeholders and that also parents should be involved in it with the help of a multidisciplinary team, including also psychologists.

Round 2

Reviewer 2 Report

Comments and Suggestions for Authors

Dear Authors,

the quality of the study is really poor. In my opinion the results bring nothing new to the field.

Author Response

Comment:

The quality of the study is really poor.  In my opinion the study does not add anything new to the field.

Response:  We have been through the paper and highlighted the ways in which it contributes to the field.